# Low-Cost and Eco-Friendly Calcium Oxide Prepared via Thermal Decompositions of Calcium Carbonate and Calcium Acetate Precursors Derived from Waste Oyster Shells

**DOI:** 10.3390/ma17153875

**Published:** 2024-08-05

**Authors:** Somkiat Seesanong, Chaowared Seangarun, Banjong Boonchom, Nongnuch Laohavisuti, Wimonmat Boonmee, Somphob Thompho, Pesak Rungrojchaipon

**Affiliations:** 1Office of Administrative Interdisciplinary Program on Agricultural Technology, School of Agricultural Technology, King Mongkut’s Institute of Technology Ladkrabang, Bangkok 10520, Thailand; ksesomki@yahoo.com; 2Material Science for Environmental Sustainability Research Unit, School of Science, King Mongkut’s Institute of Technology Ladkrabang, Bangkok 10520, Thailand; chaowared@gmail.com; 3Municipal Waste and Wastewater Management Learning Center, School of Science, King Mongkut’s Institute of Technology Ladkrabang, Bangkok 10520, Thailand; 4Department of Chemistry, School of Science, King Mongkut’s Institute of Technology Ladkrabang, Bangkok 10520, Thailand; pesak.ru@kmitl.ac.th; 5Department of Animal Production Technology and Fishery, School of Agricultural Technology, King Mongkut’s Institute of Technology Ladkrabang, Bangkok 10520, Thailand; 6Department of Biology, School of Science, King Mongkut’s Institute of Technology Ladkrabang, Bangkok 10520, Thailand; wimonmat.bo@kmitl.ac.th; 7Faculty of Pharmaceutical Sciences, Chulalongkorn University, 254 Phayathai Road, Patumwan, Bangkok 10330, Thailand; somphob.t@chula.ac.th

**Keywords:** oyster shell, renewable resource, environmental issue, thermal decomposition, calcium oxide

## Abstract

Waste oyster shells were utilized to produce calcium carbonate (CaCO_3_) by grinding. This CaCO_3_ was then reacted with acetic acid to yield calcium acetate monohydrate (Ca(CH_3_COO)_2_·H_2_O). Both CaCO_3_ and Ca(CH_3_COO)_2_·H_2_O were used as precursors for synthesizing calcium oxide (CaO) through thermal decomposition at 900 °C and 750 °C, respectively. The yields of CaO from both precursors, determined through calcination experiments and thermogravimetric analysis (TGA), exceeded 100% due to the high purity of the raw agents and the formation of calcium hydroxide (Ca(OH)_2_). X-ray fluorescence (XRF) analysis revealed a CaO content of 87.8% for CaO-CC and 91.5% for CaO-CA, indicating the purity and contamination levels. X-ray diffraction (XRD) patterns confirmed the presence of CaO and minor peaks of Ca(OH)_2_, attributed to moisture adsorption. Fourier-transform infrared (FTIR) spectroscopy identified the vibrational characteristics of the Ca-O bond. Scanning electron microscopy (SEM) showed similar morphologies for both CaO-CC and CaO-CA, with CaO-CA displaying a significant amount of rod-like crystals. Based on these results, calcium acetate monohydrate (CA) is recommended as the superior precursor for synthesizing high-purity CaO, offering advantages for various applications.

## 1. Introduction

Thailand, a Southeast Asian country, is known for tropical beaches; therefore, the eastern and southern coasts of Thailand are suitable areas for various shellfish farming activities, and the bloody cockle, green mussel, and oyster shells are the first three shellfish extensively cultured in Thailand [1]. According to the statistical data of the Statistics of Marine Shellfish Culture Survey 2019 reported by Fishery Statistics Group (Fisheries Development Policy and Planning Division, Department of Fisheries, Ministry of Agriculture and Cooperatives, Thailand), the production of oysters in Thailand from 2010 to 2019 increased from 10,758 to 17,903 tons [1]. After the edible part of the oyster is consumed, lots of waste oyster shells are then generated. The main chemical component of oyster shells is calcium carbonate (CaCO_3_, more than 96 wt%) [2], which is an unlimited source that happens every day from living things that can be used to replace natural calcium ores (limited source) such as limestone [3]. Consequently, this high amount of calcium content makes oyster shells a potential starting material for the synthesis of various value-added calcium compounds.

Waste oyster shells have been applied in several applications; for example, they could be used as a precursor to synthesize other calcium compounds such as calcium hydroxide (Ca(OH)_2_) [4] and calcium hydrogen phosphate (CaHPO_4_) [5]. In addition, it could be applied in biomedical applications since oyster shells could be used as a precursor to synthesize calcium hydroxyapatite [Ca_10_(PO_4_)_6_(OH)_2_] for application as grafts for bone substitution, augmentation, or repair due to their similarity in composition to bone [6]. Oyster shells could also be employed as a precursor to synthesize a sorbent to capture the green-house CO_2_ gas [7] and to synthesize microcrystalline foams for use as thermal insulators with excellent properties: chemical and non-flammability inertness, non-toxicity, freeze resistance, and low thermal conductivity [8]. Oyster shells could be used as a stabilizer to improve soil quality [9], as an arsenic immobilizer in highly contaminated soils, as a material for soil regeneration [10], as a calcium-enriched supplement for livestock, and as a substitute material for limestone applying as a bio-cement for the construction industry [11,12]. Oyster shells could also be applied for the removal of phosphorus, algae [13], and phosphate [14] in rivers. These removal processes are beneficial for eutrophication control [15]. The eutrophication phenomenon is one of the problems threatening the mariculture industry. Enrichment by nutrients, particularly phosphorus elements, leads to blooms of various algae that interfere with aquatic animals; therefore, the removal of excess phosphates from water is an essential technique of the eutrophication control process [15].

CaCO_3_, an insoluble compound, is normally found in calcite, aragonite, vaterite, and dolomite minerals. It could be applied as a drying agent and as an additive in ceramic and glass [4]. Ultrafine ground CaCO_3_ particles (94 nm) were prepared from carbonate minerals using a low rotation speed in a planetary ball mill [16]. The CaCO_3_, suspended in emulsion, could be applied as an efficient antimicrobial material to reduce the Staphylococcus aureus (S. aureus) quantity [17]. It has been used in food industries to trap the aromatic molecules of some foods and to keep the aroma of foods for longer periods [18]. Oyster shell CaCO3 powder was applied as an adsorbent for heavy metal adsorption, such as cadmium, lead, and copper, in aqueous solutions [19]. Chiou et al. reported the transformation of three crystalline forms for oyster shell powder, namely CaCO_3_, calcium oxide (CaO), and Ca(OH)_2_ [3]. The fresh crystalline form of oyster shell powder was CaCO_3_. When the oyster shell CaCO_3_ was calcined at 900 °C, the calcined crystalline form, CaO, was then obtained; however, ambient humidity (H_2_O) can be adsorbed on the CaO surface, resulting in the formation of Ca(OH)_2_ [3]. This hydrate form, Ca(OH)_2_, has been reported as a material for improving water quality [20]. In addition, it has been applied in eco-friendly green calcium hydroxide nano-plates (GCHNPs) for industrial, chemical, and environmental applications as adsorbents and catalysts [21].

Calcium acetate is composed of a calcium cation (Ca^2+^) and two acetate anions (CH_3_COO−). Calcium acetate monohydrate (Ca(CH_3_COO)_2_·H_2_O) [22] and calcium acetate hemihydrate (Ca(CH_3_COO)_2_·0.5H_2_O) [23] are the most available forms of calcium acetate. Furthermore, anhydrous form (Ca(CH_3_COO)_2_) of calcium acetate was also reported [22]. This anhydrous form could be prepared from the thermal decomposition of Ca(CH_3_COO)_2_·H_2_O. Calcium acetate can be synthesized by the reaction between vinegar (or acetic acid, CH_3_COOH) and CaCO_3_, which is derived from different calcium sources of non-living things such as carbonate rocks, limestone, and marble [22]. In addition, it could also be synthesized by using some components of living things as the starting materials, such as eggshells [24], black snails [25], and littleneck clams [26]. Calcium acetate has been employed as an adsorbent to adsorb the greenhouse CO_2_ gas [27], as an important controller to reduce the emissions of toxic gasses in the coal combustion process such as nitrogen oxide, sulfur dioxide, and other toxic gasses [28], and as an alternative deicer applied in the industries [29]. In addition, calcium acetate has been employed to treat or prevent calcium deficiency and hyperphosphatemia in patients [30]. It has been employed to precipitate milk fat globule membrane proteins from a dairy by-product [31]. It has been employed as the precursor to synthesize other value-added compounds such as calcium phosphate [32], acetone [33], ceramic composites [34], calcium sulfate [35], and cement mortar [36]. Moreover, it has been employed as a soil pH adjuster, a foliar fertilizer, a plant micronutrient, and a soil amendment [37].

CaO, commonly known as burnt lime or quicklime, can be prepared by the thermal decomposition process of CaCO_3_-based materials such as limestone [38] and shells [39] by using a temperature above 900 °C. CaO demonstrates two dominant applications. It has been used as a low-cost solid-state catalyst for biodiesel production [40] and used as an adsorbent for CO_2_ capture [41]. Furthermore, CaO can be used in various applications, including as a filler in feed, as a fertilizer, as a bio-catalyst, in paper, printing ink, pharmaceutical and cosmetic industries, and as a starting material for the preparation of dielectric compounds such as calcium silicate (CaSiO_3_), calcium titanate (CaTiO_3_), calcium aluminate (CaAl_2_O_4_), and calcium sulfate or gypsum (CaSO_4_) [42]. Therefore, using natural waste from living organisms as a renewable source for value-added material preparation can help reduce environmental issues. The aim of this work is to find an easy and low-cost method with no environmental impact to transform waste oyster shells into CaO compounds via processing to obtain different precursors, leading to specific properties that are advantageous for specific applications such as catalysis, ceramics, and chemical production, and making it the best choice for zero-waste management. Consequently, two precursors (calcium carbonate, CaCO_3_, and calcium acetate monohydrate, Ca(CH_3_COO)_2_·H_2_O) transferred from oyster shell wastes were studied to prepare calcium oxide (CaO). The advantages of preparation of CaO powders from two precursors will be reported in the details of physicochemical properties and safety of cost and energy, which reveal some different properties of the obtained product when using different raw materials that are very important in scientific and industrial fields.

## 2. Materials and Methods

### 2.1. Preparations

The waste oyster shells used in this study were collected from Sriracha district (13.1678, 100.9204), Chonburi province, Thailand. The waste shells were first cleaned with distilled water and sodium hypochlorite (NaOCl) to remove dirt and residual tissue, respectively. The obtained waste was rinsed repeatedly with distilled water to ensure the cleaning process and then dried in an oven at 60 °C for 2 days. The cleaned shells were manually crushed and sieved using mesh No. 50 to obtain the fine CaCO_3_ (CC) powders, which were then used as raw material to synthesize calcium acetate compounds. Industrial-grade acetic acid (99.85 wt% CH_3_COOH, Merck/Burlington, MA, USA) was used as the reagent for the preparation of the calcium acetate compound. Firstly, concentrated acetic acid (99.85 wt%) was diluted with deionized (DI) water to prepare 50 wt% acetic acid. Diluted acetic acid was then slowly added into a beaker containing CaCO_3_ in the mole ratio of CaCO_3_:acetic acid of 1:2. The mixture was stirred continuously at 100 rpm using a magnetic stirrer until the exothermic reaction completed by monitoring CO_2_ gas. The synthesis reaction finished when no CO_2_ was produced. The reacted product was kept in an oven at 60 °C for 6 h, forming calcium acetate granular. This dried product was manually crushed and then sieved (100 mesh) to obtain fine Ca(CH_3_COO)_2_·H_2_O (CA) powders. The chemical reaction between oyster shell CaCO_3_ and CH_3_COOH for the synthesis of calcium acetate monohydrate is presented in Equation (1).
CaCO_3_(s) + 2CH_3_COOH(aq) ⟶ Ca(CH_3_COO)_2_·H_2_O(s) + CO_2_(g)(1)

After that, the obtained CC and CA samples were investigated by thermogravimetric analysis (TGA), which revealed calcined temperature separately at 900 and 750 °C for 1 h using a crucible. After calcination, they were mechanically milled in a ball mill machine by using a rotation speed of 60 rpm to obtain the fine CaO powders, which were labeled as CaO-CC and CaO-CA, when CaCO_3_(CC) and Ca(CH_3_COO)_2_·H_2_O (CA) were used as the precursor for the synthesis of CaO, respectively.

### 2.2. Characterizations

All prepared samples (CaCO_3_, Ca(CH_3_COO)_2_·H_2_O, CaO-CC, and CaO-CA) were characterized by the physicochemical analyzers. Thermal decomposition behaviors of two precursors (CaCO_3_(CC) and Ca(CH_3_COO)_2_·H_2_O (CA)) were analyzed by the thermal analyzer (TG/DTA Pyris Diamond, PerkinElmer, Waltham, MA, USA) via the thermogravimetric analysis (TGA) technique. Thermal decomposition investigation was conducted by using calcined alpha-alumina (α-Al_2_O_3_) powder (Sigma-Aldrich, St. Louis, MO, USA) as the reference from 30 °C to 900 °C at a heating rate of 10 °C/min under nitrogen gas (N_2_) with a flow rate of 50 mL/min [41]. Using this technique, the thermogravimetric (TG) and its derivative (or differential thermogravimetric, DTG) curves were measured and recorded. X-ray powder diffractometer (XRD, Bruker AXS, Billerica, MA, USA) was used to identify the crystal structure of the samples. To confirm the synthesized samples, the resulting XRD patterns were compared with the Powder Diffraction File (PDF) databases of the International Centre for Diffraction Data (ICDD) [43]. A two-theta (2θ) angle range from 5 to 60° (0.01° increment) and a scan speed of 1 s/step [40] were used during the XRD measurement. To confirm the vibrational characteristics of the functional group that existed in the samples, the vibrational spectra of the samples were observed and recorded by a Fourier-transform infrared (FTIR) spectrophotometer (Spectrum GX, PerkinElmer). KBr powder (spectroscopic grade) was used to mix with each sample with a quantity ratio of approximately 10:1 [44]. The mixture was finely homogenized in an agate mortar to protect the contamination, put into a sample mold, and compressed with a hydraulic compressor using a pressure of 1000 psi for 2 min, then forming a mixture as a pellet [44]. The infrared spectra were recorded in the range of 4000–400 cm^−1^ (2 cm^−1^ resolution [45], 32 number of scans).

To achieve the highest worthiness for the production of CaO products, the yields of the production processes of the CaO-CC and CaO-CA samples were investigated by thermal decomposition by thermal gravimetric analysis (TGA) and calcination of two precursors in a furnace. The total reactions of thermal decomposition for CC and CA to produce CaO powder are according to Equations (2) and (3), in which 20 g of a raw agent in the crucible was calcined by a furnace for 1 h. The percentage yield (%Yield, P_P_) of CaO production was calculated by Equation (4).
900 °CCaCO_3_ (s) → CaO (s) + CO_2_ (g)(2)
750 °CCa(CH_3_COO)_2_·H_2_O (s) → CaO (s) + CO_2_ (g) + 2CH_3_COO (g) + H_2_O (g)(3)
(4)$$P_{p}=\frac{m_{obs}}{m_{theo}}\times 100\%$$
where *m_obs_* is the obtained mass of CaO-CC or CaO-CA powders after calcination, and *m_theo_* is the theoretical mass of CaO calculated according to reactions (2) and (3).

An X-ray fluorescence (XRF, SRS 3400, Bruker, Billerica, MA, USA) spectrometer was used to investigate the elemental composition of the synthesized CaO-CC and CaO-CA samples. To prepare the XRF sample, an agate mortar was used to crush and homogenize the sample manually. Before the XRF operation, the crushed sample was pressed into a pellet using starch as a binder [46,47,48,49,50,51,52,53,54]. The surface morphologies of CaO-CC and CaO-CA samples were observed by the scanning electron microscope (SEM, VP1450, LEO, North Billerica, MA, USA). Each sample was first coated with gold powders using the sputtering technique to eliminate the electric charge that is generated in a nonconducting sample when it is scanned by a high-energy electron beam [48,49,50,51,52,53,54]. The SEM measurement was conducted at a magnification of 20,000 times.

## 3. Results and Discussion

### 3.1. CaO Production Results

The calcined temperatures of CaO production for CC and CA precursors prepared from oyster shell wastes were 900 and 750 °C, respectively. Masses of CaO-CC and CaO-CA products were observed and calculated according to quantitative chemical analysis (Equations (2) and (3)), and their production yields were determined by Equation (4). After calcination, masses of CaO-CC and CaO-CA powders were found to be 11.0250 and 6.4550 g, which are close to theoretical values of 11.2000 and 6.5116 g, respectively. The %yield values of production for CaO-CC and CaO-CA were calculated by Equation (4) and found to be 98.44 and 99.13%, respectively. The % lost masses were calculated to be 44.88 and 67.73% for the calcination of CC and Ca precursors, respectively.

### 3.2. Thermogravimetric (TGA) Analysis

The TGA technique was used to investigate the thermal decomposition behaviors of the CaCO_3_ and Ca(CH_3_COO)_2_·H_2_O samples. The thermal decomposition processes of samples were measured in the temperature range of 30–900 °C. Figure 1a demonstrates the TG and DTG curves of the CaCO_3_ sample. As can be seen in the TG-DTG curves, the single thermal decomposition step was observed with a temperature range of 630–880 °C. The corresponding DTG peak at 846 °C with the DTG value of −0.160 mg/min was assigned as the elimination of CO_2_ gas. This obtained thermal decomposition temperature agrees with the result reported by Aqliliriana et al. [48]. They conducted the TGA processes of the limestone and mud creeper shell and reported that limestone and mud creeper shell were thermodecomposed completely at 800 and 900 °C, respectively. Furthermore, the mass-loss percentage of 42%, according to the TG result obtained in this work, also confirmed the decomposition of CO_2_ gas from the CaCO_3_ particles, resulting in the formation of CaO products [40]. Equation (2) can be used to present the thermal decomposition of oyster shell CaCO_3_ powder. In theoretical thermal analysis for 100% purity of CaCO_3_, the % mass loss and %mass remains need to be 44 and 56%, respectively. This result is consistent with the % purity and the presence of contamination of the CaCO_3_ obtained from oyster shells, which was revealed by XRF techniques. Additionally, the % mass remains (58%) is corresponding to the %yield of CaO production from the calcination of CaCO_3_ obtained from oyster shells.

The thermal decomposition of the synthesized Ca(CH_3_COO)_2_·H_2_O compound was also investigated, and the corresponding decomposition curves are presented in Figure 1b. The TG curve of the Ca(CH_3_COO)_2_·H_2_O presented three significant thermal transformation processes with the temperature ranges of 75–187 °C, 334–535 °C, and 542–723 °C with the corresponding three mass-loss percentages of 12%, 31%, and 23%, respectively. The total % mass loss of 66% and the % mass remains of 34% agree with those in theoretical values (67.44 and 32.56%) and are consistent with the observed %yield of CaO production from the calcined CA sample.

The first TG mass-loss step of the synthesized Ca(CH_3_COO)_2_·H_2_O consisted of two first DTG peaks, and the number of these two DTG peaks indicated the unlike surrounding environment of the water molecules. The first two DTG peaks at 117 and 166 °C were assigned as the elimination of the weak and strong interactions of water molecules in the structure that existed in the crystal structure of the sample (Ca(CH_3_COO)_2_·H_2_O). The elimination of water in the latter case was called the “dehydration” process. The elimination of the water molecule in the crystal structure created an anhydrous form of calcium acetate or Ca(CH_3_COO)_2_. This result is in agreement with the result observed by Bette et al. [22]. They calcined the Ca(CH_3_COO)_2_·H_2_O at 300 °C, and the Ca(CH_3_COO)_2_ was then obtained.

The second TG mass-loss step consisted of a DTG peak at 424 °C and its shoulder at around 497 °C. This second thermal decomposition is called a “complex process” due to the overlapping characteristics between two or more two thermal reactions, such as a DTG peak and its shoulder. More detail was described in the literature [51,55,56]. The DTG peak at 424 °C was assigned as the decomposition of Ca(CH_3_COO)_2_, resulting in the formation of CaCO_3_ together with the elimination of acetone (CH_3_COCH_3_). This thermal decomposition was then called the “deacetonation” process. The shoulder-like DTG peak at 497 °C was assigned as the thermal decomposition of acetone, resulting in the formation of ketene (H_2_CCO) and methane (CH_4_). These ketenes and methane were then thermoeliminated, which were called the “deketenation and demethanation” processes [55]. After that, the final TG mass-loss step with a DTG peak at 694 °C was assigned as the decomposition of CaCO_3_ with the elimination of CO_2_, which was called the “decarbonation” process, resulting in the formation of CaO. Consequently, the thermal decomposition mechanism of the synthesized calcium acetate hydrate could be described as the following equations (Equations (3), (5), (6), and (2)). This thermal analysis result reveals optimized temperature to produce acetone (CH_3_)_2_CO at 424 °C), ketene (H_2_CCO at 497 °C), and methane(CH_4_ at 497 °C) from Ca(CH_3_COO)_2_·H_2_O agent prepared by oyster shell and acetic acid reaction, which are different from those values from the same compound prepared by different starting agents and synthesis methods [55,56,57].

Dehydration (first and second DTG peaks: 117 and 166 °C) is described by Equation (3).

Deacetonation (third DTG peak: 424 °C)
Ca(CH_3_COO)_2_(s) ⟶ CaCO_3_(s) + CH_3_COCH_3_(g)(5)

Deketenation and demethanation (fourth shoulder-like DTG peak: 497 °C)
CH_3_COCH_3_(g) ⟶ H_2_CCO(g) + CH_4_(g)(6)

Decarbonation (fifth DTG peak: 694 °C) is described by Equation (2).

### 3.3. X-ray Fluorescence (XRF) Results

The chemical compositions of CaO-CC and CaO-CA were revealed by X-ray fluorescence (XRF) spectroscopy, and the resulting compositions are listed in Table 1. The most abundant component of CaO-CC and CaO-CA were 87.80% CaO and 91.50% CaO, respectively. Furthermore, both the CaO-CC and CaO-CA samples contained small amounts of various oxide compounds and some halogens, e.g., chlorine and bromine elements. The experimental XRF data indicated that CaO-CA shows a higher calcium content (3.7%) with lesser impurities (8.5246%) than that of CaO-CC. This result also confirmed that, compared to CC, CA should be selected first as the precursor to synthesize CaO. Additionally, contamination composition may affect the %yield, which supports the observed %yield of CaO production from two precursors obtained from oyster shell wastes.

### 3.4. X-ray Diffraction (XRD) Results

The crystal structures of the CC, CA, CaO-CC, and CaO-CA samples were investigated using the X-ray diffraction technique. The diffraction patterns of the synthesized CC and CA are shown in Figure 2a and Figure 2b, respectively. The XRD pattern, as shown in Figure 2a, pointed out and confirmed the diffraction characteristics of calcium carbonate (CaCO_3_) with the calcite polymorph according to the PDF #47-1743 database [48]. The diffraction positions (2θ) at 29.48° and 47.58° are the major characteristic of calcite CaCO_3_ polymorph, which corresponded to hkl planes of (104) and (018), respectively. The calcite CaCO_3_ crystals crystallize in a trigonal crystal system [45] with the lattice parameters of a = b = 4.9763 Å and c = 17.0904 Å, the corresponding unit cell volume of 366.520 Å^3^, and the space group of R-3c (space group #167) [50]. The atomic bond lengths of the crystal structure of CaCO_3_ were investigated and reported by Antao et al. [51]. They observed that the bond length between carbon and oxygen (C–O) atoms is 1.2836 Å, whereas the bond length between calcium and oxygen (Ca–O) atoms is 2.3574 Å.

The crystallography of the synthesized calcium acetate compounds was also investigated, and the resulting XRD pattern is presented in Figure 2b. The experimental XRD result was compared with the XRD patterns reported by Lee et al. [24], Park et al. [25], and Dionysiou et al. [55]. They synthesized calcium acetate using different techniques, and the diffractometer was then employed to characterize the products. When compared to the previous works [24,25,56,57,58], the XRD pattern of the synthesized calcium acetate obtained in this work is in agreement with the XRD pattern of calcium acetate monohydrate (Ca(CH_3_COO)_2_·H_2_O). Furthermore, the XRD pattern of the Ca(CH_3_COO)_2_·H_2_O obtained in this work is also in good agreement with the reference data given by Klop et al. [56] and van der Sluis [57]. The ICDD database of PDF #19-0200 also confirms that the product synthesized from the reaction between the oyster shell CaCO_3_ and CH_3_COOH is the Ca(CH_3_COO)_2_·H_2_O [55]. Klop et al. [56] investigated the crystal structure of Ca(CH_3_COO)_2_·H_2_O and described that Ca(CH_3_COO)_2_·H_2_O consists of infinite multiple oxygen-bridged double-stranded calcium chains and the gap between each calcium chain is cross-linked via the hydrogen bonds. The lattice parameters of triclinic Ca(CH_3_COO)_2_·H_2_O crystal are a = 6.751 Å, b = 11.077 Å, and c = 11.783 Å, whereas the lattice angles are α = 116.50°, β = 92.41°, and γ = 97.32°, respectively. It crystallizes in the space group of P1 (space group #1), Schoenflies symbol of C11, and the density value (D) of 1.506 g/cm. The unit cell volume is 777.1 Å^3^, and the number of formula units in a unit cell (Z) is 4 [53,56,59].

Figure 3a and Figure 3b show the diffraction patterns of CaO-CC and CaO-CA, respectively. The diffraction results of CaO-CC and CaO-CA show the same characteristics and have two important peaks corresponding to the (111) and (220) reflections (hkl planes) at 2θ (diffraction positions) of 32.24° and 54.44°, respectively. The existence of these peaks shows the characteristics of the CaO compound with PDF #00-004-0777 [41]. As demonstrated in Figure 3, the diffractograms of CaO-CC and CaO-CA are similar, indicating that CaCO_3_ and Ca(CH_3_COO)_2_·H_2_O have thermally been changed to CaO completely. The diffraction patterns of the prepared CaO-CC and CaO-CA samples are in good agreement with the diffraction pattern of CaO material reported by Aqliliriana et al. [48]. They prepared cubic CaO crystals from the thermal decomposition of limestone and mud creeper shell precursors. These precursors were subsequently thermodecomposed at 800 °C and 900 °C, resulting in the formation of cubic CaO [56] with the space group of Fm-3 m (space group #225), lattice parameters a = b = c = 4.8152(3) Å, and the unit cell volume of 111.64 Å^3^ [53,56,59].

However, as demonstrated in both Figure 3a and Figure 3b, the diffraction peaks of Ca(OH)_2_ were also observed due to the PDF#01-070-5492 [53,56,59]. Based on this PDF database, the diffraction positions (2θ) at 18.08°, 28.76°, 34.18°, 47.22°, and 50.90°, which are identified to the (001), (100), (101), (012), and (110) reflections, respectively. These characteristic peaks of Ca(OH)_2_ are in good agreement with the result reported by Chanda et al. [60]. They synthesized the nanocrystalline portlandite (Ca(OH)_2_) with the hexagonal phase from the chemical reaction of calcium nitrate (Ca(NO_3_)_2_), cetyl trimethyl ammonium bromide (CTAB, C_19_H_42_BrN), and sodium hydroxide (NaOH) reagents. The Ca(OH)_2_ commonly forms when CaO is exposed to the atmosphere conditions [60,61]. The atmospheric moisture (H_2_O) existent in the air was adsorbed on the surface of CaO, forming Ca(OH)_2_ with the space group of p-3m1 (space group #164), lattice parameters a = b = 3.59694 Å and c = 4.91861 Å, and unit cell volume of 55.08 Å^3^ [60,61]. Consequently, this result supports the reason for the %yield of the CaO production, which was discussed before.

### 3.5. Fourier-Transform Infrared (FTIR) Results

An FTIR spectrophotometer was used to investigate the vibrational characteristics of CaCO_3_ (CC) and Ca(CH_3_COO)_2_·H_2_O (CA) prepared from oyster shells and CaO products prepared from CC and CA. The resulting infrared adsorption spectra are demonstrated in Figure 4 and Figure 5. The FTIR spectrum of oyster shell powder is demonstrated in Figure 4a. The spectroscopic curve shown in this figure confirmed that the chemical composition of the oyster shell powder is CaCO_3_ due to the presence of the vibrational characteristics of carbonate (CO_3_^2−^) anion [29,59,61,62]. Some vibrational peaks were then explained. The broad bands in the range of 3672–3022 cm^−1^ are the symmetric and asymmetric stretching modes of both the O–H of water (H_2_O) and the C–H of methyl (–CH_3_) groups. The two peaks at 2919 and 2511 cm^−1^ are the combination and overtone types of the C–O symmetric and C–O asymmetric stretching modes of CO_3_^2−^ [40,61]. The peak at 1794 cm^−1^ is the C=O stretching mode, whereas the peak at 1421 cm^−1^ is the C–O asymmetric stretching mode of CO_3_^2−^. The peak at 874 cm^−1^ is the C−O symmetric stretching mode, whereas the peak at 709 cm^−1^ is the out-of-plane and in-plane bending of CO_3_^2−^ [40,61,62].

The FTIR spectrum of the synthesized CA sample is demonstrated in Figure 4b. Some vibrational peaks were then explained. The broad bands in the range of 3687–2949 cm^−1^ are the C–H symmetric and C–H asymmetric stretching modes of the methyl (–CH_3_) group [22]. In addition, this broadband region is also assigned to the O–H symmetric and O–H asymmetric stretching modes of H_2_O. The band in the range of 1680–1561 cm^−1^ is the H–O–H symmetric bending mode of H_2_O [22,24]. The band in the range of 1561–1487 cm^−1^ is the C–O asymmetric stretching mode, while the band in the range of 1487–1361 cm^−1^ is the C–O symmetric stretching mode of the acetate anion (CH_3_COO–). The band in the range of 1077–972 cm^−1^ is the out-of-plane stretching mode of the CH_3_ group, whereas the band in the range of 972–919 cm^−1^ is the C–C stretching mode of CH_3_COO− [22,24]. The band in the range of 701–601 cm^−1^ is the out-of-plane O–C–O stretching mode of CH_3_COO−. Finally, the three peaks in the range of 499−398 cm^−1^ are the in-plane O–C–O bending mode of CH_3_COO− [22].

Figure 5a and Figure 5b show the FTIR spectra of CaO-CC and CaO-CA products, respectively. The wide and strong band at around 500 cm^−1^ corresponds to the vibrational characteristic mode of the Ca–O bond [62]. The vibrational characteristics of the C–O stretching mode of carbonate (CO_3_^2−^) groups were observed at the absorption bands of 1416 cm^−1^ and 872 cm^−1^ for CaO-CC and 1419 cm^−1^ and 875 cm^−1^ for CaO-CA. In addition, the sharp peaks at 3636 cm^−1^ for CaO-CC and 3637 cm^−1^ for CaO-CA are assigned as the O–H stretching mode of water physisorbed on the CaO surface [63,64], which is in good agreement with the OH group in the crystal structure of Ca(OH)_2_ as shown and described in the XRD results. These vibrational peaks (3636 and 3637 cm^−1^) also confirmed that when the synthesized CaO was exposed to atmospheric moisture, Ca(OH)_2_ could form from the adsorption process of moisture (H_2_O) by CaO. The FTIR result is consistent with the XRD results to explain the obtained %yield for CaO production previously discussed above.

### 3.6. The Scanning Electron Microscope (SEM) Results

The morphologies of CaO-CC and CaO-CA observed by the scanning electron microscope (SEM) are demonstrated in Figure 6a and Figure 6b, respectively. The SEM process was conducted at a magnification of 20,000 times. The morphology of CaO-CC (Figure 6a) displays irregular particles of polyhedral of small (0.1–0.3 μm width and 0.5–1.0 μm length) and large 0.4–0.7 μm width and 0.6–1.0 μm) sizes, which are agglomerates with rough surface. The microstructure of CaO-CA (Figure 6b) shows many rod-like crystals with small (0.1–0.2 μm width and 0.5–1.0 μm length) and large 0.3–0.4 μm width and 0.6–5.0 μm) sizes. The morphologies of CaO-CC and CaO-CA differ in shape and size, which could be affected by different temperatures of calcination and evolved gas kind of thermal decomposition in each step. Overall characterization results confirm that oyster shell CaCO_3_ powders were successfully employed as a renewable resource to synthesize calcium acetate compound (Ca(CH_3_COO)_2_·H_2_O), and CaO was obtained when CaCO_3_ and Ca(CH_3_COO)_2_·H_2_O were thermodecomposed. As mentioned in the Introduction section regarding potential applications, future work will focus on investigating the performance of the synthesized CaO (CaO-CA and CaO-CA) for use as a low-cost catalyst for biodiesel production [40] and as an adsorbent for CO_2_ capture [41].

## 4. Conclusions

This work presented experimental data for CaO production using biowaste (waste oyster shells) as a renewable resource through a low-cost and eco-friendly method. The waste oyster shells were transferred to two precursors, CaCO_3_(CC) and Ca(CH_3_COO)_2_·H_2_O (CA), which were then calcined at 900 and 750 °C to obtain the final CaO products, affecting the cost and some physical properties to apply for specific use. The XRD and FTIR results confirm the phase of two precursors (CaCO_3_ and Ca(CH_3_COO)_2_·H_2_O) and the final decomposed CaO products obtained from the same starting agent, oyster shells. The %yield, %purity, and contamination of the prepared CaO-CC and CaO-CA products are different, which may affect some applications such as catalyst use. The morphologies of CaO-CC and CaO-CA observed from the scanning electron microscope exhibited shapeless and rod-like particles, indicating different sizes and shapes, which are caused by the CaO products obtained from different precursors. This work reports an easy and low-cost method with no environmental impact to transform waste oyster shells into CaO compounds via processing to obtain different precursors, leading to specific properties for targeted uses, making it the best choice for zero-waste management. The result of this research is very advantageous for specific applications such as catalysis, ceramics, and chemical production.

## Figures and Tables

**Figure 1 materials-17-03875-f001:**
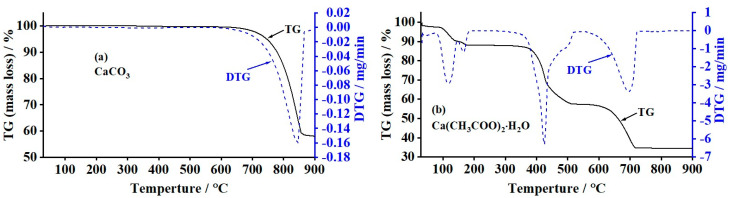
Thermogravimetric analyses (TG and DTG curves) of (**a**) CaCO_3_ and (**b**) Ca(CH_3_COO)_2_·H_2_O compounds obtained from oyster shells.

**Figure 2 materials-17-03875-f002:**
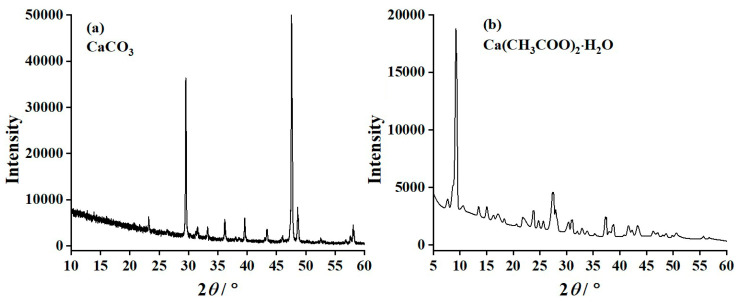
X-ray diffraction (XRD) patterns of (**a**) CC and (**b**) CA products derived from oyster shell wastes.

**Figure 3 materials-17-03875-f003:**
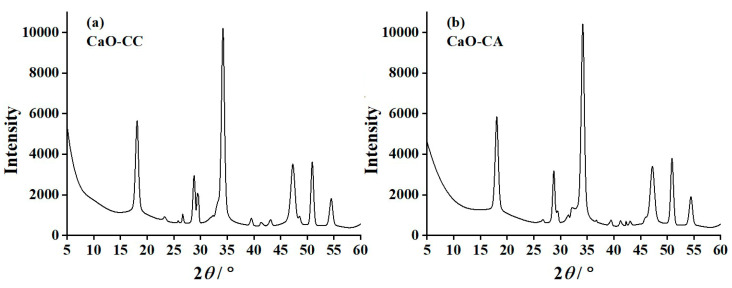
X-ray diffraction (XRD) patterns of (**a**) CaO-CC and (**b**) CaO-CA.

**Figure 4 materials-17-03875-f004:**
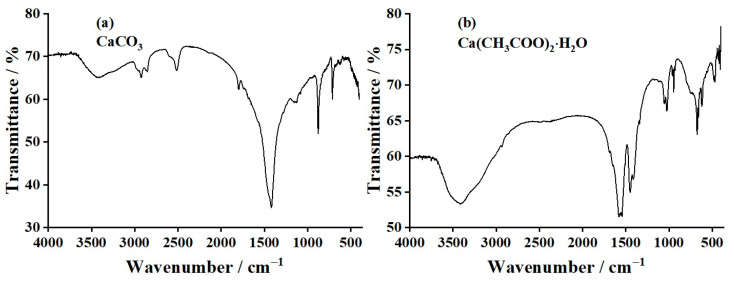
Fourier-transform infrared (FTIR) spectra of (**a**) CaCO_3_ (CC) and (**b**) Ca(CH_3_COO)_2_·H_2_O (CA) products prepared from oyster shells.

**Figure 5 materials-17-03875-f005:**
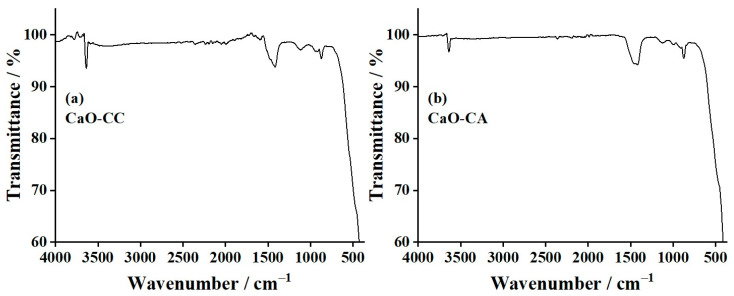
Fourier-transform infrared (FTIR) spectra of (**a**) CaO-CC and (**b**) CaO-CA.

**Figure 6 materials-17-03875-f006:**
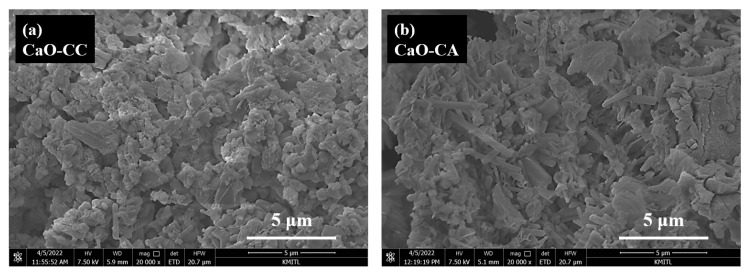
Scanning electron microscopic (SEM) images of CaO products prepared from the thermal decomposition of (**a**) CaCO_3_ and (**b**) Ca(CH_3_COO)_2_·H_2_O derived from oyster shells.

**Table 1 materials-17-03875-t001:** Chemical compositions of CaO products prepared by the calcination of CaCO_3_(CaO-CC) and Ca(CH_3_COO)_2_·H_2_O (CaO-CA) derived from oyster shells, investigated by XRF technique.

Chemical Compositions		Chemical Contents/wt%		
		CaO-CC	CaO-CA	Differences
Calcium oxide	CaO	87.80	91.50	3.70
Sodium oxide	Na_2_O	0.83	0.38	0.45
Magnesium oxide	MgO	2.08	1.46	0.62
Aluminium oxide	Al_2_O_3_	1.10	0.75	0.35
Silicon dioxide	SiO_2_	4.58	2.86	1.72
Phosphorus oxide	P_2_O_5_	0.21	0.19	0.02
Sulfur dioxide	SO_3_	1.10	0.97	0.13
Chlorine	Cl	0.79	0.61	0.18
Potassium oxide	K_2_O	0.21	0.16	0.05
Titanium dioxide	TiO_2_	0.10	0.08	0.01
Manganese oxide	MnO	0.06	0.05	0.01
Ferric oxide	Fe_2_O_3_	0.81	0.65	0.16
Cupric oxide	CuO	0.01	0.01	-
Zinc oxide	ZnO	0.01	0.01	-
Bromine	Br	0.01	0.01	-
Strontium oxide	SrO	0.30	0.31	0.01
Total impurities		12.1639	8.52	
Total		100.00	100.00	

## Data Availability

The data are contained within the article.

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
