# Peer review of "Low-Cost and Eco-Friendly Calcium Oxide Prepared via Thermal Decompositions of Calcium Carbonate and Calcium Acetate Precursors Derived from Waste Oyster Shells"

_materials, 2024, doi:10.3390/ma17153875_

Round 1

Reviewer 1 Report

Comments and Suggestions for Authors

    Oyster shells were employed in the manuscript as raw materials to prepare calcium oxide by two different pathways, including direct calcination and transformation calcination. The experimental setting is clear, and the analysis and testing methods used are conventional and feasible. Several issues need to be considered as follows.

1. The literature has been well reviewed but the main purpose of the present work is not addressed, and the purpose of the prepared samples is unclear. The specific target leads to the necessary tests and analyses。

2.  What is the purpose of calculating the Pp value in Eq. 4? Due to the presence of impurities in the raw materials, the calculated P-value is higher than 100%, which is not an unexpected or interesting value as reported in lines 225-230.

3. Section 3.2, there is doubt about the analyses of TG and DTG curves in Fig1b. In general, it states that the decomposition temperature of calcium acetate is above 160 degrees Celsius. The authors addressed it on 424 degrees Celsius, and the reasons need to be explained. Furthermore, as described in the manuscript, the TG/DTG curves track the weight loss of acetone (gas phase?) decomposition at 497 degrees Celsius?  Is that possible? Further tests are needed to prove its accuracy.

4.The post-processing of XRD curves were incomplete and cannot accurately locate characteristic peaks to illustrate their composition characteristics.

5.The post-processing of FTIR curves were incomplete and cannot accurately locate characteristic peaks to illustrate their composition characteristics.

6. Finally, The Supplementary Materials files mentioned in the article are not available

Author Response

Reply to the reviewer

Comments and Suggestions for Authors

    Oyster shells were employed in the manuscript as raw materials to prepare calcium oxide by two different pathways, including direct calcination and transformation calcination. The experimental setting is clear, and the analysis and testing methods used are conventional and feasible. Several issues need to be considered as follows.

  1. The literature has been well reviewed but the main purpose of the present work is not addressed, and the purpose of the prepared samples is unclear. The specific target leads to the necessary tests and analyses。

Answer:  The aim of this work is finding an easy and low-cost method with no environmental effect to transform waste oyster shells to CaO compound via processing to obtain different precursors lead to get specific properties, which is advantageous for specific applications such as catalysis, ceramics, and chemical production and the best choice for zero waste management. Consequently, two precursors (calcium carbonate, CaCO3, and calcium acetate monohydrate, Ca(CH3COO)2·H2O) transferred from oyster shell wastes were studied to prepare calcium oxide (CaO). The advantages of preparation of CaO powders from two precursors will be reported in the details of physicochemical properties and safety of cost and energy, which reveal some different properties of the obtained product when using different raw materials that are very important in scientific and industrial fields.

  1. What is the purpose of calculating the Pp value in Eq. 4? Due to the presence of impurities in the raw materials, the calculated P-value is higher than 100%, which is not an unexpected or interesting value as reported in lines 225-230.

Answer:  the production yields were rechecked by reperformance and then the  revised manuscript was rewritten.

  1. Section 3.2, there is doubt about the analyses of TG and DTG curves in Fig1b. In general, it states that the decomposition temperature of calcium acetate is above 160 degrees Celsius. The authors addressed it on 424 degrees Celsius, and the reasons need to be explained. Furthermore, as described in the manuscript, the TG/DTG curves track the weight loss of acetone (gas phase?) decomposition at 497 degrees Celsius? Is that possible? Further tests are needed to prove its accuracy.

Answer: - The first two DTG peaks at 117 and 166°C were assigned as the elimination of the weak and strong interactions of water molecules in the structure that existed in the crystal structure of the sample (Ca(CH3COO)2·H2O).

-The second TG mass-loss step consisted of a DTG peak at 424 °C and its shoulder at around 497 °C. This second thermal decomposition is called a “complex process” due to the overlapping characteristics between two or more two thermal reactions such as a DTG peak and its shoulder.

-This thermal analysis result reveals optimized temperature to produce acetone (CH3)2CO at 424 oC ), ketene (H2CCO at 497 oC) , and methane(CH4 at 497 oC)  from Ca(CH3COO)2·H2O agent prepared by oyster shell and acetic acid reaction.

Deacetonation (3rd DTG peak: 424 °C)

Ca(CH3COO)2(s) ⟶ CaCO3(s) + CH3COCH3(g)                               (7)

4.The post-processing of XRD curves were incomplete and cannot accurately locate characteristic peaks to illustrate their composition characteristics.

Answer: The authors don't agree with the comment and sure that the XRD curves appropriate.

5.The post-processing of FTIR curves were incomplete and cannot accurately locate characteristic peaks to illustrate their composition characteristics.

Answer: The authors don't agree with the comment and sure that the FTIR curves appropriate.

  1. Finally, The Supplementary Materials files mentioned in the article are not available

Answer:  the sentence Supplementary Materials: The following supporting information can be downloaded at: www.mdpi.com/xxx/s1, Figureure S1: title; Table S1: title; Video S1: title. ”  was deleted from the revised manuscript.

Reviewer 2 Report

Comments and Suggestions for Authors

The article is interesting and concerns the use of waste oyster shells to obtain products beneficial for the economy. Such research is important in the context of the circular economy and should be published.

Before publication, please address the following comments.

In abstract, the use of "%" before "yields" is not necessary.

The introduction is well-written and contains valuable information about the usefulness of the target products. The objective is concisely stated. However, there is a lack of emphasis on the research gap.

Line 181, what do 32 scans mean? Is the final scan an average of 32 scans?

Line 190 3 should be in index

Line 192. Reaction is not correct. You should check the hydrogen balance. The dots on the left side of the equation are unnecessary.

Line 195-196 obs and theo should be in index

Are lines 207-210 the content of the manuscript?

It is not possible to achieve yields above 100%, and such data should not be published. The cause of such a result should be investigated using additional analyses. Assumptions are not sufficient. Were repetitions performed? If so, please add information about the statistics."

What was the gas atmosphere during the TG analysis? Will the samples decompose the same way in an air atmosphere or an inert gas atmosphere?

Line 271 Can the authors explain the occurrence of weak and strong interactions of water molecules in the structure? "Is it about chemically bound water in the structure and water as moisture?

Line 292 The authors write that 'result reveals optimized temperature to produce acetone, ketene, and methane.' What type of optimization was applied?"

In reactions, subscript 'g' indicates the gaseous state of the compound. The arrows are redundant.

Line 399 and 414, What does Flourier mean?

What does the strongly decreasing transmittance below 500 cm¹ indicate

Can the authors provide additional information on the size, shape and distribution of particles in the microstructure of CaO-CC and CaO-CA? Are there specific morphological features that may influence their physical and application properties?

The conclusions are too general and should be revised. Emphasize the quantitative results

What are the key challenges to overcome before scaling?

Author Response

Reply to the reviewer

The article is interesting and concerns the use of waste oyster shells to obtain products beneficial for the economy. Such research is important in the context of the circular economy and should be published.

Answer: Thank you very for your kind appreciation in this

Before publication, please address the following comments.

In abstract, the use of "%" before "yields" is not necessary.

Answer:  the authors agree with the reviewer so "%" before "yields"  was deleted from the abstract.

The introduction is well-written and contains valuable information about the usefulness of the target products. The objective is concisely stated. However, there is a lack of emphasis on the research gap.

Answer:  the authors agree with the reviewer so

Line 181, what do 32 scans mean? Is the final scan an average of 32 scans?

Answer: “32 scans” were replaced by 32 number of scans.

Line 190 3 should be in index

Answer:  CaCO3

Line 192. Reaction is not correct. You should check the hydrogen balance. The dots on the left side of the equation are unnecessary.

 Answer:           Ca(CH3COO)2.H2O(s).  → CaO(s)+CO2(g)+2CH3COO + H2O(g)                  (3)

Line 195-196 obs and theo should be in index

Answer:   mobs and mtheo

Are lines 207-210 the content of the manuscript?

Answer:  Thank you for your helping and these sentences were deleted from the revised manuscript as below:

Research manuscripts reporting large datasets that are deposited in a publicly available database should specify where the data have been deposited and provide the relevant accession numbers. If the accession numbers have not yet been obtained at the time of submission, please state that they will be provided during review. They must be provided prior to publication.

     Interventionary studies involving animals or humans, and other studies that require ethical approval, must list the authority that provided approval and the corresponding ethical approval code.

It is not possible to achieve yields above 100%, and such data should not be published. The cause of such a result should be investigated using additional analyses. Assumptions are not sufficient. Were repetitions performed? If so, please add information about the statistics."

Answer:  The authors agree with the reviewer, the production yields were reperformed and  then were rewritten in the revised manuscript.

What was the gas atmosphere during the TG analysis? Will the samples decompose the same way in an air atmosphere or an inert gas atmosphere?

Answer:   an inert gas atmosphere (under nitrogen gas (N2) with a flow rate of 50 mL/min) 

Form the decomposition reaction of CaCO3 and Ca(CH3COO)2.H2O, the evolved gases are CO2(g) and CH3COO(g) and H2O(g), which do not react with the N2 gas                               

Line 271 Can the authors explain the occurrence of weak and strong interactions of water molecules in the structure? "Is it about chemically bound water in the structure and water as moisture?

Answer:  the production yields were rechecked by reperformance and then the sentence was deleted from the revised manuscript.

Line 292 The authors write that 'result reveals optimized temperature to produce acetone, ketene, and methane.' What type of optimization was applied?"

Answer:  the type of optimized temperature to produce acetone, ketene, and methane from calcium acetate obtained from this work are applied at 424 oC ,  497 oC, 497 oC, respectively.

In reactions, subscript 'g' indicates the gaseous state of the compound. The arrows are redundant.

Answer:  The arrows were deleted from the revised manuscript.

Line 399 and 414, What does Flourier mean?

Answer:  Thank your for your kind suggestions and the Flourier were changed to Fourier.

What does the strongly decreasing transmittance below 500 cm⁻¹ indicate

Answer:  the strongly decreasing transmittance below 500 cm-1 because the chemical compound does not have fundamental vibrational frequency and the limit of detection by this method is the range of 4000-400 cm-1, which is near the end.

Can the authors provide additional information on the size, shape and distribution of particles in the microstructure of CaO-CC and CaO-CA? Are there specific morphological features that may influence their physical and application properties?

Answer:  The morphology of CaO-CC (Figure. 6a) displays irregular particles of polyhedral of small (0.1-0.3 mm width and 0.5-1.0 mm length) and large 0.4-0.7 mm width and 0.6-1.0 mm) sizes, which are agglomerates with rough surface. The microstructure of CaO-CA (Figure. 6b) shows many rod-like crystals with small (0.1-0.2 mm width and 0.5-1.0 mm length) and large 0.3-0.4 mm width and 0.6-5.0 mm)sizes.  The morphology of CaO-CC  and CaO-CA are different shape and sizes, which could be affected by different temperatures of calcination and  evolved gas kind of the thermal decomposition in each step.

The conclusions are too general and should be revised. Emphasize the quantitative results

Answer:  the conclusions were rewritten.

 What are the key challenges to overcome before scaling?

Answer:  The aim of this work is finding an easy and low-cost method with no environmental effect to transform waste oyster shells to CaO compound via processing to obtain different precursors lead to get specific properties, which is advantageous for specific applications such as catalysis, ceramics, and chemical production and the best choice for zero waste management. Consequently, two precursors (calcium carbonate, CaCO3, and calcium acetate monohydrate, Ca(CH3COO)2·H2O) transferred from oyster shell wastes were studied to prepare calcium oxide (CaO). The advantages of preparation of CaO powders from two precursors will be reported in the details of physicochemical properties and safety of cost and energy, which reveal some different properties of the obtained product when using different raw materials that are very important in scientific and industrial fields.

Reviewer 3 Report

Comments and Suggestions for Authors

the paper is extremely interesting and presents a new  approach to prepare calcium derivatives from waste materials.

the introduction is complete and informative as for the topic. the experimental section is complete and the reactions and analytical determinations have been described in detail. The discussion is exhaustive.

only some typos to be corrected.

Comments on the Quality of English Language

the paper is extremely interesting and presents a new  approach to prepare calcium derivatives from waste materials.

the introduction is complete and informative as for the topic. the experimental section is complete and the reactions and analytical determinations have been described in detail. The discussion is exhaustive.

Author Response

Reply to the reviewer

Comments and Suggestions for Authors

the paper is extremely interesting and presents a new  approach to prepare calcium derivatives from waste materials.

the introduction is complete and informative as for the topic. the experimental section is complete and the reactions and analytical determinations have been described in detail. The discussion is exhaustive.

Answer: Thank you very for your kind appreciation in this

only some typos to be corrected.

Answer: the revised manuscript has been rechecked.

Comments on the Quality of English Language

the paper is extremely interesting and presents a new  approach to prepare calcium derivatives from waste materials.

the introduction is complete and informative as for the topic. the experimental section is complete and the reactions and analytical determinations have been described in detail. The discussion is exhaustive.

Answer: Thank you very for your kind appreciation in this

Round 2

Reviewer 1 Report

Comments and Suggestions for Authors

No more comment